

# A Review of MIS 5e Sea-level Proxies around Japan

Evan Tam[1,2] and Yusuke Yokoyama[1,2,3,4,5]

[1]Atmosphere and Ocean Research Institute, The University of Tokyo. 5-1-5 Kashiwanoha, Kashiwa, 277-8564, Japan
[2]Graduate Program on Environmental Sciences, Graduate School of Arts and Sciences, The University of Tokyo. 3-8-1 Komaba, Meguro-ku, Tokyo, 153-8902, Japan.
[3]Department of Earth and Planetary Science, Graduate School of Science, The University of Tokyo. 7-3-1 Hongo, Bunkyo-ku, Tokyo, 113-0033, Japan.
[4]Biogeochemistry Program, Japan Agency for Marine-Earth Science and Technology. 2-15 Natsushima-cho, Yokosuka-city, Kanagawa, 237-0061, Japan.
[5]Research School of Physics, The Australian National University. Canberra, ACT 2601, Australia.

*Correspondence to*: evan.tam@g.ecc.u-tokyo.ac.jp

**Abstract.** Sea-level proxies for Marine Isotopic Stage 5e (MIS 5e, ca. 124 ka) are abundant along the Japanese shoreline, and have been documented for over at least the last 60 years. The bulk of these sea-level proxies are identified in Japan as marine terraces, often correlated by stratigraphic relationships to identified tephra layers, or other chronologically interpreted strata. Use of stratigraphic correlation in conjunction with other techniques such as paleontological analysis, tectonic uplift rates, tephra (volcanic ash), Uranium-Thorium (U/Th), Carbon-14 ($^{14}$C), and Optically Stimulated Luminesce (OSL) dating techniques have connected Japan's landforms to global patterns of sea-level change. This paper reviews over 60 years of publications containing sea-level proxies correlated to forming during MIS 5e in Japan. Data collected for this review have been added to the World Atlas of Last Interglacial Shorelines (WALIS), following their standardizations on the elements necessary to analyze paleo sea-levels. This paper reviewed over 70 studies, assembling data points for 300+ locations and examining related papers denoting sea-level indicators for MIS 5e. The database compiled for this review (Tam and Yokoyama, 2020) is available at: https://doi.org/10.5281/zenodo.4294326.

## 1    Introduction

Marine Isotope Stage (MIS) 5e is of particular interest because of its position as the last major interglacial period before present, and due to similarities in global mean temperatures during this period to projected changes in climate, observations of MIS 5e could aid in quantifying sea-level change in the current and coming century (Stirling et al., 1995; Rohling et al., 2008; Rahmstorf, 2007; Church et al., 2001). This stage has been constrained to between 128–116 ka (Stirling et al., 1998; Yokoyama and Esat, 2011), with average sea-level rise in tectonically stable areas at 5–9 m higher than at present (Dutton and Lambeck, 2012). Sea-level increases are credited to warmer global temperatures, an increased influx of icebergs into the ocean, or varying degrees of both (Overpeck et al., 2006; Otto-bliesner et al., 2006; Yokoyama and Esat, 2011). Accurate measurements of changes in ocean basin sea water volume and ice sheet volume is necessary to parameterize the effects of tectonics, isostasy, and eustacy on fluxes in sea-level for a given location (Milne, 2014; Yokoyama et al., 2018; Yokoyama et al., 2019a). This data is vital for accurate GIA modeling and calculation of tectonic uplift rates using coastal sea-level proxies (Okuno et al., 2014; Fukuyo et al., 2020).



This paper serves as context to the data collected on MIS 5e sea-level proxies in and around Japan. The
database was compiled as a part of the World Atlas of Last Interglacial Shorelines (WALIS), which aims to
globally compile MIS 5e sea-level indicators in a standardized format (https://warmcoasts.eu/world-atlas.html).
Descriptions of each database fields can be found here: https://doi.org/10.5281/zenodo.3961543 (Rovere et al.,
2020), compiled at the following website: https://walis-help.readthedocs.io/en/latest/. The regional database for
sea-level indicators of Japan during this period can be found at the following link:
https://doi.org/10.5281/zenodo.4294326 . This database reviewed over 70 studies, extracting 315 representative
sea-level indicators across Japan. Among these, 310 proxies were age constrained by stratigraphic correlation,
149 utilized tephra–stratigraphic correlation, 6 used OSL dating, and 5 employed U/Th dating, with studies
frequently using multiple techniques.

## 2 Literature Overview

### 2.1 Geologic Background

The Japanese Archipelago is tectonically one of the most active locations in the world (Ando et al.,
2018; Nakanishi et al., 2020; Yokoyama et al., 2016), consisting of several island arcs created by the collision of
at least 5 plates: the Amurian, Eurasian, Okhotsk, Pacific, and Philippean Sea Plates (Figure 1). The archipelago
is primarily composed of 4 large islands: Hokkaido, Honshu, Shikoku, and Kyushu. Subduction of the Pacific
Plate beneath the Okhotsk, and the Philippine Sea Plate, form the Kuril and Izu–Ogasawara arcs and the
Northeast Honshu arc. Additionally, subduction of the Philippine sea plate beneath the Okhotsk and Amurian
plates form the Southwest Honshu arc, and subduction beneath the Eurasian plate form the Ryukyu arc (Taira,
2001; Moreno et al., 2016; Apel et al., 2006). This unique convergence of plates results in distinct uplift and
subsidence patterns that alter marine terrace elevations (Ota and Omura, 1991), with MIS 5e created sea-level
indicators at elevations ranging from -85.5 m to 205 m. Japan is also host to a large number of active volcanoes
due to its tectonic activity, and records of volcanic activity are vitial in constraining ages of terraces and sea-
level proxies (as discussed later).

Almost all studies of sea-level proxies defining sea-level maxima during MIS 5e in Japan utilize
analyses of marine terraces. Relatively high uplift rates are found in many coastal regions in Japan, preserving
sea-level highstands as staircase terraces. Terraces have been previously subcategorized into 3 types:
topographically defined marine terraces, sedimentologically defined marine terraces, and terraces defined by
paleontological evidence (Ota and Omura, 1991), though many studies provide little information marine terrace
details. Sea-level proxies in this study are categorized according to definitions provided in Rovere et al.., 2016
(Table 1).

### 2.2 Historical Studies and U/Th Dating

Earlier studies chronicling sea-level proxies in Japan generally utilized paleontological evidence to
constrain marine deposit ages (Kamada and Niino, 1955; Sakaguchi, 1959; Yonekura, 1968), or used proxies to
calculate Quaternary crustal movement (e.g., Yoshikawa, 1964; Ota, 1971). Marine terraces were correlated to
the Riss–Würm Interglacial period (then identified between 90–100 ka), but have since been reassessed to align
with MIS 5e sea-level highstands. Paleontological proxies such as mollusca species were utilized to identify



warmer climate conditions associated with the deposition of sea-level highstand marine sediments (e.g., Yonekura, 1968).

In the 70's, the utilization of Uranium–Thorium (U/Th, aka U-series) dating provided age constraints on fossilized coral terraces representing sea-level highstands globally. Since then, studies examining Kikai Island and other Ryukyu islands combined with results from Barbados (Thompson et al., 2011; James et al., 1971) and Papua New Guinea (Chappell, 1974; Chappell et al., 1996; Yokoyama et al., 2001a, b), have reconfirmed constrained dates of terraces representing sea-level highstands, matching age groups of approximately 120, 100, 80, and 60 ka (Konishi et al., 1974, Yonekura et al., 2001). Ages from oxygen isotope analyses of deep sea sediment cores also corresponded to these high sea-level periods, linking these analyses together and more accurately defining MIS 5e, 5c, and 5a (Lisiecki and Raymo, 2005; Yokoyama et al., 2019a; Ota, 1986). Though U/Th dating continues to be used in Japan (e.g., Inagaki and Omura, 2006), suitable samples of carbonate origin are generally found only in the Ryukyu islands (Ota and Omura, 1991).

### 2.3 Chronostratigraphy and Tephrochronology

Studies examining sea-level proxies in Japan heavily rely on chronostratigraphic correlations, employing key widespread tephra and stratigraphic layers, the latter of which are often constrained by the former. Machida (1975)'s use of tephrochronology with fission track dates allowed for correlation to high sea-level stages as observed in Papua New Guiniea and Barbados (Ota and Omura, 1991; Chappell, 1974; Chappell et al., 1996; Yokoyama et al., 2001a, b), paving the way for the use of tephra and pumice layers as a common chronohorizon dating technique in stratigraphic analysis. Characterization of glass mineral assemblages and chemical composition through electron microprobe, instrumental neutron activation analysis, and inductively coupled plasma mass spectrometry has allowed for identification of chemical signatures of specific tephra layers, linking these layers to specific eruptive events and volcanoes (Machida, 2002). Thus, it became possible to link widely distributed tephra layers and associated stratigraphic layers/marine terraces by age and to Marine Isotope Stages (Machida and Arai, 2003). Key tephra layers from individual eruptions have broad distributions, with Japanese sourced tephra layers identified in Korea and the Ryukyu Islands (Figure 2; Machida, 2002).

Dating of tephra layers is essential in constraining ages of stratigraphic layers, and [14]C, fission-track, U/Th, thermo-luminescence, electron spin resonance, and K-Ar dating techniques have all been utilized to establish and cross–check ages associated with tephra depositional events. Due to the wide distribution of tephra layers and the plethora of dating techniques available for analyzing them, chronostratigraphic correlation to identified tephra layers or age constrained stratigraphic layers is considered reliable and heavily relied upon in Japan (Machida, 2002; see Table 4).

Of the many tephra layers identified and employed as reliable chronostratigraphic horizons, the Toya tephra, Zarame Pumice (ZP), Aso-4 and Aira-Tn (AT) layers have broad distributions and are commonly used to constrain ages of sea-level proxies around Japan (Figure 2). The Toya tephra is widely distributed over much of Hokkaido and northern Honshu, sourced from eruptions that formed the Toya Caldera (Machida, 1987). Ages have been constrained to between 112–115 ka by stratigraphic correlations of tephras and terrace heights (Machida, 2002), though Zircon U-Th-Pb dating and aliquot regeneration–red thermal luminescence dating have given ages of 108±19 ka and 104±30 –118 ± 30 ka (Ito, 2014; Ganzawa and Ike, 2011). The ZP layer was





deposited as thick airlaid tephra from unknown volcano, and is found below the Toya tephra and above MIS 5e

surfaces in stratigraphic sequences, with ages estimated by Miyauchi (1985) between 110–120 ka (Matsuura, 2019; Miyauchi, 1989). The ZP layer has been idenfied in studies examining northern Honshu, though mainly in the well studied Kamikita Coastal region where middle and late pleistocene terraces are widely distributed on multiple levels (e.g., Matsuura, 2019).

        The AT tephra is one of the most widespread tephra in Japan, with traces having been found in

Kyushu, Shikoku, Honshu, and Korea (Machida and Arai, 2003, 1983; Machida, 2002). The tephra was sourced from three phases of eruptions of the Aira caldera in northern Kagoshima Bay, and has been dated by $^{14}$C to an age of 25.12±0.27 BP (Miyairi et al., 2002; Machida and Arai, 2003). The Aso-4 tephra layer represents the youngest and largest tephra layer from the Aso Caldera in central Kyushu and was distributed as far as eastern Hokkaido, making it ideal for terrace chronology (Machida, 2002; Aoki 2008). Ages between 86.8–87.3 ka were

obtained from detailed $\delta^{18}$O isotopic stratigraphy from ocean cores collected in the northwest Pacific Ocean and the Sea of Okhotsk (Aoki, 2008).

        Though techniques defining tephra ages have become more precise over time, over–reliance on tephra based chronostratigraphy can be precarious, as certain tephra layers have been and still are described with large age uncertainties. Although the Toya tephra has since been more accurately constrained (Ito, 2014; Gannzaka

and Ike, 2011), historical utilizations of ages from the original fission track age, along with ages from stratigraphic constraints of the ash layer in the northern part of Japan resulted in a range of 90–130 ka (Okumura and Sagawa, 1984; Miyauchi, 1988; Ota and Omura, 1991). An applied example, updated tephra defined marine terraces ages from Tanegashima (Machida et al., 2001) compared to the original age interpretation (Ota and Machida, 1987) show a discrepancy of 20 ka. Large error ranges from various dating techniques combined with

tephra layer ages defined solely by stratigraphic correlation alone indicate that while tephrostratigraphy is viable, direct dating of tephra layers and sea-level proxies should be utilized when available. It should also be noted age correlation of sea-level proxies in the absence of tephra layers is not uncommon when deemed equivalent to well constrained proxies within the region (e.g. Koike and Machida, 2001).

### 2.4    Tectonic Uplift Studies

140        The reliance on tephrochonology based chronostratigraphy and chronostragraphic correlation without use of a direct dating technique highlights the frequent lack of directly datable samples associated with sea-level proxies in Japan. Marine terraces, when found without reliable tephra layers, have been correlated by counting interglacial deposits/terraces backwards from MIS 5e (Ito et al., 2017), or by comparing relationships within a series of terraces where one terrace constrained by recognized tephra layers, $^{14}$C dating (for younger terraces in

the series), or paleontological proxies such as mollusks (Koike and Machida, 2001). Due to the relationship between uplift and terrace preservation, regional uplift rates have been utilized to assign MIS stages to terrace sequences, and likewise terrace ages have been used to calculate uplift rates.

        Often, studies that identify sea-level proxies in Japan focus on calculating regional tectonic uplift rates and patterns (e.g., Suzuki et al., 2011; Miyazaki and Ishimura, 2018). As uplift rate calculations require dating

of an uplifted proxy, MIS 5e terraces can be utilized for their defined age range. When possible, absolute dating techniques (see below for more techniques) or tephrochronology are utilized to constrain ages to calculate the



regional uplift rates (e.g., Hiroki, 1994; Ota and Odagiri, 1994). If direct dating techniques cannot be employed, stratigraphic correlation has been relied on to constrain terrace ages. Terrace heights, regional uplift rates, and sediments signifying transitions between sea-level highstands have been used to designate MIS 5e terraces in

lack of datable material (Yoshikawa et al., 1964), as have river profiles to sea-level/marine terrace height relationships (Yoshiyama, 1990) and implied regional stratigraphic relationships (Koike and Machida, 2001). However, these techniques introduce a higher possibility of dating errors due to the use of stratigraphic relationships (as mentioned earlier) and age calculation based on regional uplift rates, which rely on the assumption of constant uplift over the proxy's history.

**2.5    Other Techniques**

While the abovementioned techniques represent the bulk of techniques commonly utilized in sea-level proxy identification, many others have been employed in assessing their ages. In addition to U/Th dating, tephrochronology, and stratigraphic correlation, a limited number of studies utilizing OSL dating have been performed on marine terraces in Japan. Samples from the Noto peninsula, the Kamikita coast, and the Oga

peninsula analyzed utilizing thermoluminesce and multiple–aliquot additive dose (MAAD) quartz OSL dating (Tanaka et al., 1997), K–feldspar post–infrared infrared (pIRIR) stimulated luminsence dating (Ito et al., 2017), and both quartz OSL and K–feldspar pIRIR dating (Thiel et al., 2015), respectively. Results from Thiel et al. (2015), and Ito et al. (2017), suggest that K–feldspar pIRIR dating is appropriate for dating marine terraces and marine sediments formed during MIS 5 and older, even in locations where quartz OSL is deemed unsuitable.

Limited studies have utilized cosmogenic nuclide dating ($^{10}$Be and $^{26}$Al) to analyze MIS 5e and MIS 7 associated terraces in the Kii Peninsula and Shikoku (Yokoyama et al., 2015, 2019b). Amino acid racemization has seen limited use in constraining MIS 5e terrace ages in Japan (e.g., Ota and Odagiri, 1994), as has electron spin resonance (ESR) dating (e.g., Ikeya and Omura, 1983).

**3    Database details**

As a part of this review, over 70 papers, including 3 databases and the references therewithin were examined. Direct latitude and longitude values were provided only in limited studies (specifically in databases provided in Pedoja et al., 2011; Pedoja et al., 2014), so locations were estimated by comparing mapped locations provided in published studies to Google Earth, or finding an appropriate average location for areas examined in the study. Due to the large quantities of data examined (in Koike and Machida (2001) alone 2000+ data points),

this review aims to broadly represent studies conducted throughout Japan.

**3.1    Data Collection and Calculations**

Sea-level proxy elevations and uplift rates were recorded from data sources when values were clearly articulated in reviewed studies or could be interpreted from figures. Data retrieved from Koike and Machida (2001) was averaged for each given location. Elevation and uplift rate values describing a single location were

summed and divided by the total number of utilized values for the average elevation and uplift rate, and noted within our database as averaged. Data from other studies were added to the database to be representative of each region.

Few examined studied listed sea-level proxy elevation Margin of Error (MoE) values, so values were assigned based on the measurement technique utilized as described in Rovere et al., 2016 (Table 2). For proxies



that had elevations averaged from multiple points, half of the range between the highest and lowest proxy
elevations was added to the MoE. Sea-level proxies with large ranges in elevation resulted in rather large MoEs,
which are denoted in the RSL quality rating as less reliable (see Section 5.1).

Tidal ranges were calculated for the Japanese coastline to calculate Indicative Range (IR), Relative Water
Level (RWL), and the Upper and Lower Limits (UL, LL; as defined in Rovere et al., 2016) for modern analogs

of sea-level proxies. Tidal predictions were provided by Hydrographic and Oceanographic Department of the
Japan Coast Guard (2020). Tidal predictions for all functional tide gauges were examined for dates between
January 1st, 2020 to March 31st, 2020, to calculate the average, maximum, and minimum sea-level height, and
sea-level range for each day, and the overall examined time period. The Japanese coastline was divided into 59
sectors, based on similarities in tidal changes during this period [Appendix A].

For marine terraces and beach deposits, IR and RWL with the data and formulas of IMCalc (Lorscheid and
Rovere, 2019). Instead of the standard tidal values in IMCalc, the tidal values calculated in this study were
utilized. UL and LL for coral terraces proxies or those with relevant molluscan constraints were evaluated
manually (Table 3) to reflect a more accurate sea-level range due to proxy formation below sea-level. Sea-level
extent for coral or mollusk defined proxies can reach from 0–30 m in Japan but can be further constrained by

identifying key species (Yokoyama and Esat, 2015; Nakamori et al., 1995). Coral reef habitat extent ranges
from the mean lower low water (MLLW) to the end of forereef (Rovere et al., 2016). Using the IR and RWL
obtained for each sector, UL and LL for coral terrace and mollusk constrained proxies were calculated as
follows:

$$UL = RWL - (^{IR}/_2) \tag{1}$$

$$LL = UL - MD_p \tag{2}$$

where *RWL, IR, UL, and LL* represent the relative water level, indicative range, upper limit, and lower limit, and
$MD_p$ represents the maximum depth of the proxy examined (Table 3).

Paleo sea-level and sea-level uncertainties were evaluated within the WALIS database, using the principles
outlined in Rovere et al. (2016). Paleo sea-levels for each location were calculated using the following formula:

$$RSL_p = E\text{-}RWL \tag{3}$$

where $RLS_p$ represents the paleo sea-level, *E* is the current proxy elevation, and *RWL* is the modern relative
water level. The associated MoE for each proxy was calculated with the following formula:

$$\sigma_{RSL} = [(E_e)^2 + (IR/2)^2]^{1/2} \tag{4}$$

where $\sigma_{RSL}$ is the proxy's paleo sea-level MoE, $E_e$ is the elevation MoE, and *IR* is the modern indicative range.

Paleo sea-level uncertainties are captured within σ_RSL, with *IR* of the modern analog describing the range over
which the sea-level proxy formed (Shennan, 1982; Van de Plassche, 1986; Hijma et al., 2015, Rovere et al.,
2016), and $E_e$ representing uncertainties in the elevation measurements.

Values denoted as averages within our database should be taken as overviews of the data provided for the
area and should not be used for rigorous calculations. Paleo sea-level calculations and their associated MoE do



not directly account for subsidence or uplift that has occurred over its lifetime. Proxy data points were rejected when the background references could not be evaluated or did not provide a usable elevation value.

### 3.2 Sea-level Indicators

Studies reviewing MIS 5e sea-level proxies in Japan seldom differentiate the term "marine terrace" with other types of sea-level indicators. As such, there is frequently ambiguity in how terraces are defined, especially when utilized as reference points to examine tephra layer relations or to calculate tectonic uplift rates. Terrace composition is often described in studies, but this information is not often utilized to differentiate between types of terraces. Sea-level indicators examined were categorized as marine terraces, beach deposits, and coral reef terraces, as defined in Rovere et al., 2016 (Table 1).

### 3.3 Elevation Details

Little information was provided in studies reviewed about sea-level proxy elevation datums utilized. Some studies reported elevations measured by barometric altimeter, total station or hand level, differential GPS, or from using elevations reported on topographic maps, though often the measurement technique was not reported (Table 2). The sea-level datum utilized is relative mean sea level (RLS), namely assumed to be Mean Sea-level (MSL), and does not correct for changes in sea-level due to eustacy or glacial isostatic adjustments. Uplift rate margin of errors were not reported in most studies, so procedures outlined in Pedoja et al., (2011) were utilized to calculate rates for studies that reported them. Each proxy elevation MoE was divided by 124,000 years, and reported in mm/yr. Rates were calculated relative to MSL, and likewise do not factor sea-level changes due to eustacy or glacial isostatic adjustments.

## 4 Sea-level Proxies: Regional Overview

Sea-level proxies as recorded in the WALIS database are described in the following section. These datapoints were divided into 8 regions for description and analysis based on geographic location and regional patterns as follows: Hokkaido, Northern Honshu, Kanto, the Noto Peninsula, the Kii Peninsula and Shikoku, Japan Seaside: Kansai and Chugoku, Kyushu and Yamaguchi, and the Ryukyu Islands (Figure 3). Proxy elevations range between -85.5±5 m and 205±5 m for all of Japan, and patterns in elevation changes are indicative of tectonic activity across the archipelago. Individual transects within regions can have large variations in proxy elevations (Figures 4–11), and many of the studies conducted denoting proxy elevations have utilized them to investigate tectonic uplift rates.

### 4.1 Hokkaido

Sea-level proxies in Hokkaido are numerous and have been well documented (Figure 4). In particular, Okumura (1996) reported terraces across the island, constraining proxies with their relationship to the Toya (112–115 Ka), KP-IV(115–120 Ka), Kc-Hb (115–120 Ka), ZP (110–120 Ka) and Mb-1 (> 130 Ka) tephra layers. The first three ash layers are sourced from Hokkaido volcanoes, specifically from Lake Toya Caldera (Machida et al., 1987) for the former, and from the Kutcharo Volcano for the latter two (Hasegawa et al., 2012).

Sea-level proxies in Hokkaido can be examined in 5 subcategories: northeast, southeast, northwest, southwest, and the western cape. Proxies on the northeastern edge of Hokkaido are age constrained by the Toya, KP-IV, Kc-Hb, and Mb-1 tephra layers, and are low in elevation compared to the rest of the island. Elevations generally ranging between 6±1.20–18±7.60 m. The proxies closer to the Nemuro Strait increase from



33.50±35.70–80±16 m. Proxies along the southeastern edge of Hokkaido are constrained predominantly by the
Toya and KP-IV tephras, but additionally ZP and Kc-Hb layers (Okumura, 1996; Koike and Machida, 2001;
Machida et al., 1987). Higher elevations can be found towards the center (35±7–60±12 m), and decreases
moving outwards (15±3–32.50±11.5 m). Both the northeastern and southeastern edge are described along
several transects by Okumura (1996),  and marine terraces correlated to the last interglacial period have been
designated as M1 stage terraces, which are often observed in sequence with H1, H2, and M2 terraces. M1
terraces are composed of marine sediments and overlie fluvial gravel, though no other terrace descriptions are
provided.

Elevations of sea-level proxies on the northwestern edge of Hokkaido generally range between 40±8–
45±9 m in elevation, though on the northern tip range from 40±8–61±32.20 m. Inland marine terraces near
Sapporo range from 30±6–52.50±15.50 m in elevation. Sea-level proxy ages are defined primarily through
stratigraphic correlation, though a few are directly constrained by Toya and Kc-Hb tephra layers (Koike and
Machida, 2001; Machida et al., 1987). Sea-level proxies on the western arm of Hokkaido vary between 20±1–
130±10 m in elevation, with lower elevations in areas further to the north or south (20±1–45±25 m). Proxies
toward the center of this range have elevations between 55±21–130±26 m, with the highest elevations found on
Okushiri Island. Terraces on the eastern side of the arm also have higher elevations (50±30–90±38 m). Ages are
constrained primarily by use of the Toya tephra and stratigraphic relationships (Koike and Machida, 2001;
Machida et al., 1987), though ZP layers are also found at select locations including Okushiri Island (Miyauchi,
1988). While sea-level proxies for southwestern Hokkaido have been studied (Yoshiyama, 1990), elevation
values were not specifically recorded.

### 4.2    Northern Honshu

Proxy elevations in northern Honshu can be subdivided into Mutsu Bay/Shimokita Peninsula, upper
eastern, lower eastern, and western regions (Figure 5). Marine terraces are well defined and categorized on the
eastern edge, and are recognized and named as the Fukuromachi, Shichihyaku, Tengutai, Takadate, Nejo, and
Shibayama terraces around the Kamikita Plains (upper section of the eastern region, Miyauchi, 1985, 1987;
Koike and Machida, 2001), with the sand-gravel marine deposit Takadate terrace constrained by the Toya tephra
to correlate to MIS 5e (Miyauchi, 1985; Ito et al., 2017). Proxies in northern Honshu are constrained by the
Toya tephra and the ZP layer (Miyauchi, 1988; Machida et al., 1987) and stratigraphic correlation. The
relatively detailed understanding of terrace layers and age constraints in this region has encouraged trials of
pIRIR OSL dating in this region (Ito et al., 2017, Thiel et al., 2015), establishing it as a viable dating method for
marine sediments.

Sea-level proxy elevations around Mutsu Bay itself range between 13.5±3.70–35±22 m, while further
north on the Shimokita Peninsula range between 30±6–50±10 m. Terraces are generally constrained by the Toya
tephra and the ZP layer, but Tanabu A, B, C (MIS 7–8, Matsuura et al., 2014) tephra layers have also been
identified to underlie MIS 5e terraces at certain sites. More recent studies from Matsuura et al. (2014) and
Watanabe et al. (2008) have explored marine terraces on the Shimokita peninsula in depth to examine regional
tectonic uplift and deformation.



Terraces on upper eastern side of northern Honshu have elevations between 15±0.08–51±2 m overall, though generally range between 35–45 from the bottom of the Shimokita Peninsula down towards Hachinohe. Between Hachinohe and Kuji, sea-level proxy elevations vary between 22.25±5–48.50±3 m, though most are between 25–30 m, generally decreasing towards the south. Most terraces are constrained by an observed Toya tephra layer, and in some areas by the ZP layer. OSL dates from AIST (2015, 2016) and Ito et al., (2017) for

MIS 5e terraces as examined in Matsuura et al. (2019) are noted to align with results from tephrochronology, though OSL ages from terraces representing MIS 7 and 9 from the same studies were found not to match tephrochronologically restrained ages. Sea-level proxies in this transect are identified by their beach deposit sequences, mainly silt, sand, and gravel deposits (Miyazaki and Ishimura, 2018; Miyauchi, 1985).

On the lower eastern side between Miyako and Ishinomaki, sea-level proxy elevations varied between

17.83±8.56–25.33±11.10 m. Ages were constrained through stratigraphic correlation (Koike and Machida, 2001; Miura, 1966), and the DKS tephra layer was observed in Matsuura et al. (2009). Terraces reported in Miura (1966) were initially correlated to the Shimosueyoshi interglacial period, which has since been reinterpreted as the Last Interglacial Period. Terraces reported by Matsuura et al. (2009) were described wave cut benches. Proxies south of Ishinomaki had relatively higher elevations (60±12, 67.50±18.50 m).

Sea-level proxies on the western side of northern Honshu are lower in elevation towards the northern tip (19.25±14.85–30±6 m), and drastically increase moving south (72±54.50–140±28 m). Proxy elevations decrease to 45.5±14.5–53.67±26 m in the Noshiro Plain (Miyauchi, 1988; Naito, 1977). Two locations have drastically lower elevations of 2.5±0.5 m (Thiel et al., 2015) and 21±18 m (Naito, 1977), though ages for the former were well constrained by both tephrochronology and pIRIR OSL dates. Elevations of terraces found on

the Oga Peninsula are relatively high (80±16 m, 130±26 m, Miyauchi, 1988), and proxies found south of this range from 25–45 m. Sea-level proxy heights are also found on islands along the western shoreline, including Tobishima (58.88±23.80 m), Awashima (54.55±21.93 m), and Sado Island (45.57±20.02–120±24 m). Age correlations were made through mainly the Toya and ZP tephra layers, though K-Tz and SK tephra layers were also noted (Watanabe and Une, 1985; Koike and Machida, 2001).

**4.3  Kanto**

Studies identifying sea-level proxies from the Last Interglacial in Kanto denote terraces mainly in Ibaraki and Chiba prefectures (Figure 6). Tephra utilized to constrain sea-level proxy ages are sourced predominantly from Mt. Hakone (Hk-Tp, Hk-KIP-8, Hk-KIP-7), though Miwa-L, K-Tz layers and Shimosueyoshi Loam are also utilized in this region.

Sea-level proxies in the upper part of Ibaraki prefecture (north of Hitachinaka) have elevations between 52.75±11.55–74±14.80 m decreasing towards the south, and are chronostratigraphically constrained mainly by the Miwa-L pumice layer, in addition to the K-Tz and Hk-KIP-7 layers (Suzuki, 1989). In the Joban region to the south, sea-level proxies are observed with elevations between 23.23±9.46–50±10 m, and increasing drastically south on the Boso Peninsula (maximum elevation of 130±10 m; Sugihara, 1970; Koike and Machida,

2001; Kaizuka, 1987). The tilting towards the northeast is thought to be at least partially due to uplift related to the activity of the Sagami Trough to the southwest of the Boso Peninsula (Tamura et al., 2010). Proxies are mainly constrained by the presence of Miwa-L and Hk-KIP-8 layers, in addition to Hk-Tp, On-Pm1 and the





Shimsueyoshi Loam (Suzuki, 1989; Suzuki, 1992).[14]C dating was utilized on identified mollusks (Crassotrea Gigas) to constrain a MIS 1 stage terrace and correlate other highstand related terraces accordingly at

Yokaichiba (Koike and Machida, 2001). OSL dating using quartz grains identified ages of shallow marine sediments from near Lake Kitaura, identifying sequences correlated to MIS 5e–5c (Hataya and Shirai, 2003). One sea-level proxy was denoted in Sagami Bay (160±32 m), and other locations in the bay have been studied, though did not articulate elevations (Koike and Machida, 2001; Machida, 1973).

### 4.4    The Noto Peninsula

Sea-level proxies on the Noto Peninsula itself are primarily age constrained through general stratigraphic correlation, though tephra layers are more numerously identified in locations to the east and the southwest (Figure 7). East of the peninsula, the easternmost two terrace elevations continue lower elevations seen in northern Honshu (30±6, 45±49 m), but moving west towards the peninsula elevations are higher (81.67±21.33, 85±17 m) and are age constrained by FR pumice and KT layers (Koike and Machida, 2001). On the Noto

Peninsula, the northern tip has generally higher elevations (maximum at 85.44±69.08 m), decrease drastically towards the middle of the peninsula (18.06±14.61–36.09±38.20 m), and increase proceeding south (37.62±44.52–52.55±16.51 m), aligning with the southward tilting of the peninsula observed by Ota and Hirakawa (1979). Age constraints of sea-level proxies on the Noto peninsula are mainly from stratographic correlations, though the Shimosueyoshi Loam layer has also been identified (Toma, 1974).

South of the peninsula sea-level proxy elevations range between 31.75±36.35–46±42.2 m, and increase near Fukui (67.13±115.42–117.71±42.50 m). Terrace ages are mainly constrained by DKP and AT tephra layers, especially terraces found south of Fukui, though SK and Aso-4 tephra layers have also been identified (Yamamoto et al., 1996; Koike and Machida, 2001).

### 4.5    The Kii Peninsula and Shikoku

Sea-level proxies found on the Kii Peninsula and the Shikoku Region can be subcategorized into 5 sections: eastern Kii, western Kii, Osaka Bay, eastern Shikoku, and western Shikoku (Figure 8). Age constraints for proxies in this general region are determined primarily through stratigraphic correlation, though K-Tz, AT tephra layers, and amino acid racemization dates were utilized in select studies.

        Sea-level proxy locations on the eastern side of the Kii peninsula range between 20±14–40±8 m and
utilize stratigraphic correlation to constrain ages to MIS 5e (Muto, 1989; Hiroki, 1994; Koike and Machida, 2001), though several additional locations have been reported without elevations around Mikawa Bay (Koike and Machida, 2001). Elevations on the western side of the Kii peninsula generally increase in elevation towards the south tip from 18.75±5.33–63.17±7.28 m and rely on stratigraphic correlation to MIS 5e (Yonekura, 1968; Koike and Machida, 2001). Terraces are described as one of a set of seven ($H_1$–$H_4$ and $L_1$–$L_3$, with $L_1$

representing MIS 5e), and are described as wave based erosional formed marine terraces, covered by later deposited sand and gravel layers (Yonekura, 1968). Proxies around Osaka bay exhibited elevations 34.80±26.96–59±44.80 m with lower elevations on the eastern side of the bay, and higher elevations to the north-northwestern side. Proxies on Awaji-shima are between 41.25±13.25–45±9 m, with one location constrained by AT Tephra (Koike and Machida, 2001; Machida, 2002).



Elevations of proxies studied on the eastern side of Shikoku range vary greatly between 57.80±27.60–173±34.60 m, increasing towards the southern tip (Yoshikawa et al., 1964; Yonekura, 1968; Matsuura, 2015; Mizutani, 1996; Koike and Machida, 2001). The terraces are identified by their inner edges, with boulders through fine silt as terrace deposits (Matsuura, 2015). Older studies utilize stratigraphic correlation for age constrainment, though several tephra layers including the K-Tz layer are recognized by Matsuura (2015).

Western Shikoku has a small number of evaluated proxies, with elevations ranging from 26±5.20–36.58±28.30 m (Ota and Odagiri, 1994; Koike and Machida, 2001). 5 terrace layers were identified (H$_1$–H$_3$, M, L) with the M terrace recognized as representing MIS 5e (Ota and Odagiri, 1994). Ages from shell Amino Acid Racemization of an underlying layer (ca. 138 ka) and overlying K-Tz tephra were used to constrain terrace layers ages (Ota and Odagiri, 1994; Mitsushio et al., 1989).

### 4.6    Japan Sea Side: Kansai and Chugoku

Few studies have been performed in this region identifying MIS 5e sea-level proxies (Figure 9). Two marine terraces by Wakasa bay (elevations of 40±8, 50±10 m), were age constrained from stratigraphic correlation. One submerged sea-level indicator was observed through seismic surveys of sediments in Miho bay, and identified MIS 5e associated sediment layers at a depth of -42±.08 m constrained by DMP tephra (Inoue et al., 2005). Additional locations in Kyoto, Tottori, and Shimane prefectures (Machida and Arai, 1979; Koike and Machida, 2001) have been studied, but were reported without elevation values.

### 4.7    Kyushu and Yamaguchi

Numerous sea-level proxies have been identified in Kyushu and Yamaguchi, with most elevations identified with low to negative values (Figure 10). Kyushu is a source of several key indicator tephra layers, and many sea-level proxies are well constrained by the Ata and Aso-4 layers. This region can be examined in 5 subsections: Yamaguchi, northern Kyushu, eastern Kyushu, southern Kyushu, and western Kyushu. A substantial number of sea-level proxies from around Kyushu were collected by Shimoyama et al. (1999), using molluscan fossil assemblages from both the intertidal or subtidal range to determine the marine top height.

Proxies found along the inland sea in Yamaguchi have elevations between 16.10±9.20–20.70±17.10 meters, and are age constrained by both the Aso-4 tephra layer, and stratigraphic correlation (Koike and Machida, 2001). In the northern section of Kyushu, terrace elevations between -7.5±0.40 and -8.1±0.4 m are reported, constrained by Ata tephra, in addition to terrace at 11.80±2.40 m constrained by stratigraphy. On the eastern edge of Kyushu, sea-level proxies near Oita generally range between 20.70±0.40–50±10 m in elevation, in addition to one submerged proxy (-85.50±0.40 m).Until Nobeoka, terrace elevations are between -29.90±0.40 to 18.75±6.25 m. Sea-level proxies south of this appear as both a high elevation set (74±14.8–107±0.4 m), and lower elevations (32±6.4, 33.33±16.66 m). Ages for eastern Kyushu are typically correlated between Aso-3 and Ata tephra layers, in addition to the Aso-4 layer and general stratigraphic correlation (Shimoyama, et al., 1999; Chida, 1974; Koike and Machida, 2001; Nagaoka et al., 2010).

At least 5 sea-level proxies have been identified on the southern coast of Kyushu. Terraces associated with Kagoshima bay have higher elevations (15.6±0.4 m, 52.3±0.4 m) than those on the coast (6.1±0.4 to -39±0.4 m). Elevations of proxies on the islands directly south of Kyushu are recorded at 51.5±0.4 m (Yakushima) and 120±0.4 m (Tanegashima) and are substantially higher at Tanegashima. Terraces in southern





Kyushu are all well constrained between Ata and Aso-3 tephra layers (Shimoyama et al., 1999). On the western side of Kyushu, sea-level indicators in proximity to the Ariake and Yatsushiro Seas have lower elevations

(-63.1±0.4 to 12.7±0.4 m). Proxies on Amakusa Island are comparatively higher (27.71±15.54–45±9 m), and again lower near Omura bay (7±1.4, 13.33±12.66 m). Ages are mainly constrained between Ata and Aso-3 tephras, though some locations utilize Aso-4 or stratigraphic correlation (Shimoyama et al. 1999; Kamada and Nino, 1955; Chida, 1976; Koike and Machida, 2001).

### 4.8 The Ryukyu Islands

Sea-level proxies in the Ryukyu Islands are here categorized in 3 groups: north of Okinawa, Okinawa and the Daito Islands, and west of Okinawa (Figure 11). Most sea-level indicators found in the Ryukyu Islands are coral terraces, allowing for direct U/Th dating of terraces, and $^{14}$C dating of lower terraces to constrain higher MIS highstand correlated terrace platform series. Elevations of MIS 5e correlated terraces north of Okinawa range from 43.95±46.79–66.58±103 m, aside from Kikai Island at 245±5 m. Direct dates of

corresponding terraces from U/Th were taken at Kikai Island (122.1±3.8 ka) and Tokunoshima (125±10 ka). $^{14}$C dates on Takara Island (2.3±0.15–3.3±0.13 ka) were used to correlate MIS 5e terrace dates (Koba et al., 1979; Ikeda, 1977; Inagaki and Omura, 2006; Koike and Machida, 2001).

On the Okinawa adjacent islands, elevations for sea-level proxies ranged between 23.33±14.66–55.75±21.15 m. Ages were constrained through stratigraphic correlation, and at Aguni Island younger terraces

were dated at 33.7 ka by $^{14}$C dating. On Minami and Kita Daito, elevations were measured at 12.45±2.49 m and 10±2 m, and U/Th dates were averaged at and 123±5 and 123±6 ka respectively (Omura et al., 1991; Koike and Machida, 2001; Ota and Omura, 1992). West of Okinawa, proxy elevations fell into two groups: between 11±2.2–25±15 m (Miyako, Yonaguni, Minna and Tarama Islands) and 41±8.2–60.17±46 m (Islands near Ishigaki Island). U/Th ages were calculated from coral limestones at Hateruma Island (128±7 ka), and ages were

otherwise constrained through stratigraphic correlation (Omura et al., 1994; Ota and Omura, 1992; Koike and Machida, 2001).

## 5 Further Details on Sea-level Proxies around Japan

### 5.1 Data Quality

The data quality from studies examining MIS 5e sea-level proxies in and around Japan is considered

reliable. Age constraints provided for studies are generally well supported, reporting the dating technique or rational for age assignments for specific sea-level proxies. As mentioned in section 2, use of chronohorizons can introduce larger age constraint MoEs, and can be considered suboptimal since terrace ages and MIS stages are correlated from chronostratigraphical relationships. However, due to the abundant distribution and detailed analysis of tephra in Japan, these techniques and results are considered reliable.

Details about elevation measurement styles were reported infrequent and inconsistently across the studies analyzed. Though some older studies often reported use of devices such as a Paulin Altimeter MT-2 (e.g. Yonekura, 1968), others either did not report or clearly articulate measurement styles. As a result, many studies have larger MoEs assigned to sea-level proxy elevations. Sea-level proxy type assignment similarly are infrequently delineated, with some studies listing sea-level proxies as marine terraces despite composition

details (such as coral reef terraces). Others provided little to no characteristics of the marine terraces themselves.



As explored in section 3, these details can change the interpretation of sea-level extent. Future studies could benefit from more rigorous descriptions of elevation measurement styles, sea-level proxy compositions, and details on sea-level proxy type assignments.

Data entered into the WALIS database were reported with relative sea level (RSL) and age quality ratings on a 0–5 scale rating, with 5 representing the highest value. Age quality ratings were assigned on age reliability, categorizing studies with direct dating on sea-level proxies as most reliable (5), followed by studies interpreting sea-level proxies using directly dated or constrained chronohorizons such as tephra (4), studies utilizing regional stratigraphic relationships without absolute dating to interpret sea-level proxy ages (3) , and studies using poorly described chronohorizons (2). Sea-level proxies reported in compilation databases but were deemed unverifiable due to missing source references were assigned the lowest ratings (0-1) but were omitted from this database.

RSL quality ratings were assigned using the same 1-5 scale rating. Studies were assessed on their description of sea-level proxies, including details about composition (including sediment types and identified coral/mollusk species) and sea-level proxy type assignment (such as identification of the inner margin of a marine terraces). Studies were assigned between 2-5, scaling between vague (2) to highly detailed descriptions (5). Uncertainties about sea-level proxy elevations such as rounding of elevation values of sea-level proxies were assigned low ratings (2). Sea-level proxies with elevation MoE's over 60% of the original elevation were also assigned low ratings (1). For proxies with large MoE's due to averaging, results are likely less representative of the measurement accuracy but rather indicative of the large range of sea-level proxy elevations due to regional tectonic uplift. Specific regions can have large changes in overall area, as can been seen on the Boso Peninsula (Figure 6), Sado Island (Figure 7), and Eastern Shikoku (Figure 8). Entries with the lowest rating (0) were not submitted into the database, representing data from studies that could not be located.

### 5.2 MIS 5e Sea-level Fluctuations

Overall, precise regional sea-level fluctuations analysis for the entirety of the Japanese Archipelago during MIS 5e has not been conducted, and would be difficult to constrain due to the nature of age assignments historically utilized in most studies. As use of chronohorizons and tephrochronology constrain stratigraphic layers to or between separately analyzed proxies, precise timing of sea-level changes has not emphasized in studies and can be difficult to quantify. Chronohorizons that have been themselves correlatively age constrained are common, increasing the possible margin of dating error. Studies have utilized absolute dating techniques around Japan on sea-level proxies themselves but use beyond the Ryukyu Islands has thus far been limited (e.g., Koba et al., 1979; Inagaki and Omura, 2006). Increased use of absolute dating across Japan would allow for deeper cross-regional analyses of MIS 5e sea-level fluctuations.

### 5.3 Other Sea-level Highstands

A multitude of studies have interpreted sea-level proxies in Japan to correlate to other sea-level highstands. Sea-level indicators representing MIS 11 (e.g., Hiroki, 1994), 9 (e.g., Matsuura et al., 2014, Amano et al., 2018), 7 (e.g., Ota and Omura, 1992; Miyazaki and Ishimura, 2018), 5 a–c (e.g., Inagaki and Omura, 2006; Miyauchi, 1988), and 3 (e.g., Sasaki et al., 2004; Omura et al., 2000) are abundant in the literature. Staircase terrace sea-level proxies are commonly identified, so individual studies frequently describe several



highstands or interglacial levels. Age constraints for these periods utilize and thus suffer from the same

limitations of techniques mentioned in section 2.

### 5.4    Holocene sea-level indicators

Studies on Holocene sea-level indicators are more abundant than those focusing on MIS 5e in Japan. Several reviews of Holocene sea-level changes around Japan have been compiled, such as in "Atlas of Holocene Sea Level Records in Japan" (Ota et al., 1981) and "Atlas of Late Quaternary Sea Level Records in Japan, volume 1" (Ota et al., 1987). As such, sea-level change since the last glacial maximum in Japan is well

characterized both overall and by region (Umitsu, 1991). Unlike MIS 5e sea-level proxies [14]C dating has been utilized as an absolute dating method on sea-level proxies rather than designating ages to a chronohorizon, and sees much use on Holocene emerged coral reefs (e.g., Maemoku, 1992; Hamanaka et al., 2015; Hongo and Kayanne, 2011) and on mollusks (e.g., Yokoyama et al., 2016) in the Ryukyu Islands. Largely, the same

methods for examining MIS 5e sea-level proxies are also utilized, such as tephrochronology, stratigraphic correlation, and seismic crustal analysis (e.g., Moriwaki, 2006; Nagaoka et al., 2010; Shishikura et al., 2008).

### 6    Concluding remarks

Sea-level proxies denoting MIS 5e have been abundantly observed and studied in Japan. Use of U/Th dating techniques in the coral rich Ryukyu Islands has allowed for its' terraces to be correlated to global MIS

stages, and Japan's abundant tephra sources has likewise allowed for intra-country age constraints on stratigraphic layers and identified terraces. Though chronostratigraphic techniques in Japan are recognized as reliable and accurate, opportunities exist to constrain sea-level proxies more accurately and to cross-check ages for established chronohorizons by utilizing absolute dating techniques. Several papers have validated the use of more precise absolute dating techniques on sea-level proxies in Japan, with pIRIR OSL and cosmogenic nuclide

dating techniques having been successfully used to date marine sediments and marine terraces. Future studies could benefit from more rigorous descriptions of sea-level proxy characteristics and measurement techniques utilzed. Otherwise, studies of sea-level proxies from in and around Japan have created a large, predominantly reliable collection of MIS 5e sea-level proxies that can be utilized for future studies.

### 7    Data Availability

Data from this study (Tam and Yokoyama, 2020) is open access, and available at the following link: https://doi.org/10.5281/zenodo.4294326. Data was exported from the WALIS database on 28/11/2020, and database descriptions can be found at the following link: https://doi.org/10.5281/zenodo.3961543. Further information about the database can be examined here: https://warmcoasts.eu/world-atlas.html.

*Author Contribution.* ET was the primary data compiler, analyst, and author of the paper. YY designed the research, oversaw the project, and contributed to the paper.

*Competing Interests.* No competing interests.



*Acknowledgements.* We would like to thank Alessio Rovere for inviting us to be a part of this project. Special
thanks are due to Kai Leggett, who assisted with translation, literature compilation, and data uploading, and to
Jorge Alberto Garcia Perez, Colm Murphy, and Jiwon Yeom for advising on data collection methods. This study
was supported partially by the JSPS KAKENHI grant (JP20H00193). The data compiled for this study was
uploaded into WALIS, a sea-level database interface developed by the ERC Starting Grant "WARMCOASTS"
(ERC–StG–802414), in collaboration with PALSEA (PAGES / INQUA) working group. The database structure
was designed by A. Rovere, D. Ryan, T. Lorscheid, A. Dutton, P. Chutcharavan, D. Brill, N. Jankowski, D.
Mueller, M. Bartz, E. Gowan and K. Cohen.

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



**Table 1.** Varieties of sea-level proxies identified in this study.

| Sea-level Proxy Type | Proxy Description (from Rovere et al., 2016) | Description of RWL Calculation | Description of IR Calculation |
|---|---|---|---|
| Marine Terrace | Relatively flat surfaces of marine origin, shaped by marine erosion or accumulation of sedients from erosional and depositional processes (Pirazzoli, 2005). | Tidal Prediction Heights, averaged over daily then 3 month time spans, then correlated regionally (see section 3) | Range of Tidal Prediction Heights, calculated over a daily period, then averaged over 3 month time span, then correlated regionally (see section 3) |
| Beach Deposits | Accumulations of loose sediments found on coastal surfaces, such as sand, gravel, or pebbles (Anthony, 2005). | See above | See Above |
| Coral Terrace | A marine terrace formed specifically from the interaction between bioconstructional (coral reef growth) and erosional processes (Anthony, 2008). | $MLLW - (MD_p/2)$ | $MD_p$ – see Table 3 |





**Table 2.** Elevation measurement techniques identified in this study.

| Measurement Technique | Description (from Rovere et al., 2016) | Typical Vertical Error under Optimal Conditions |
|---|---|---|
| Barometric altimeter | Difference in barometric pressure between a point of known elevation (often sea-level) and a point of unknown elevation. | Up to ±20% of elevation measurement |
| Differential GPS | GPS positions acquired in the field and corrected in real time or during post-processing. | ±0.02/±0.08 m, depending on survey conditions and instruments used |
| Metered tape or rod | The end of a tape or rod is placed at a known elevation point, and the elevation of the unknown point is calculated using the metered. | Up to ±10% of elevation measurement |
| Not reported | The elevation measurement technique was not reported, most probably hand level or metered tape. | 20% of the original elevation reported |
| Topographic map and digital elevation models | Elevation derived from the contour lines on topographic maps. Most often used for large-scale landforms (i.e. marine terraces). | Variable with scale of map and technique used to derive DEM. |
| Total station or Auto/hand level | Total stations or levels measure slope distances from the instrument to a particular point and triangulate relative to the XYZ coordinates of the base station. | ±0.1/±0.2 m for total stations, ±0.2/±0.4 m for auto or hand level. |





**Table 3.** Coral assemblage descriptions from reviewed literature used to constrain sea-level margin of error.

| Marine Assemblage | Utilized Reference | Maximum Depth of Proxy (MD$_p$) | Depth Rationale | Depth Reference |
|---|---|---|---|---|
| Coral Assemblage (No further details) | Koike and Machida, 2001 | 30 m | General Coral Range | Yokoyama and Esat, 2015; Nakamori et al., 1995 |
| Mollusca: *Mactra Sulucataria, Cycymeris vistita* | Sugihara, 1970 | 20 m | Mollusk habitat in upper shallow ocean of warm current flow | Sugihara, 1970 |
| Foraminifera: *Baculogypsina Sphaerulata, Calcarina Pengleri, Amphitegina, Lithophaga Curta, Acropora sp., Montipora sp., Goniastrea sp., Hydnophora Exesa, Symphilla Recta* | Koba et al., 1979 | 5m (Reef Crest to Upper Reef Slope) | *Baculogypsina Sphaerulata*: Range within 5 meters | Hosono et al., 2014 |
| Hermatypic Corals, Encrusting Algae, Benthic Foraminifera: *Calcarina, Baculogypsina, Marginopora* | Omura et al., 1994 | 20 m | Typical coral depth of Hermatypic corals up to 20m around Japanese Islands | Japanese Coral Reef Society, Ministry of the Environment, 2004 |
| *Crassostrea Gigas* | Miyauchi 1995 | 20 m | Intertidal to subtidal range | Harris, 2008 |
| Mollusca: *Arca Granosa L., Ostrea Palmipe Sow., Turritella cfr. Multlilirata* | Yonekura 1968, | Indicative Range | *Arca Granosa*: Intertidal zone, at 1–2 meters water depth | Pathansali, 1966 |
| Mollusca: *Paphia Undulata (Paratapes Undulatus)* | Ishii et al., 1994 | Indicative Range | Inhabits Inshore Seabed | Paphia Undulata, 2020 |
| Mollusca: *Patinepecten tokyoensis, Pecten Notovola Naganumanus, Psedoamusium Insusicostatum, Pseudoraphitoma Naganumaensis Ctuka, Mikaithyris Hanazawai* | Kamada and Nino, 1955 | 10 m | Typical *Patinopecten* habitat range is between 4–10 meters | Patinopecten Yessoensis, 2020 |
| Intertidal Molluscan Fossil Assemblage | Shimoyama et al., 1999 | Indicative Range Provided | Intertidal habitat range | |
| Subtidal molluscan fossil assemblage, including *Ophiomorpa sp.* | Shimoyama et al., 1999 | 5 m | Interpreted from *Ophiomorpha* analog *Callianassa Major*, suggested subtidal depth is 3–5 meters | Frey et al., 1978 |


**Table 4.** A list of tephra chronohorizons utilized in the reviewed literature. Modified from Machida, 2002.

| Chronohorizon Name | Abbreviation | Distribution | Dating Method Utilized | Dates (ka) | Reference |
|---|---|---|---|---|---|
| Toya | Toya | Northern Japan and surrounding oceans | OI, ST | 112–115 | Machida et al., 1987 |
| Zarame Pumice | ZP | Kamikita Coastal Plains | ST | 110–120 | Miyauchi, 1988 |
| Kutcharo Volcano | Kc-Hb | Hokkaido | FT, ST | 115–120 | Machida et al., 1987; Okumura, 1988 |
| Kutcharo Pumice Flow IV | KP-IV | Hokkaido | ST | 115–120 | Hasegawa et al., 2012; Machida et al., 1987 |
| Monbetsu Tephra | Mb-1 | Hokkaido | ST | > 125 | Okumura, 1991 |
| Daisen-Kurayoshi Tephra | DKP | Across Honshu | ST, $^{14}$C, U | 55 | Machida and Arai, 1979 |
| Kamitaru Pumice | KT | Northern Honshu | ST | 130–150 | Hayatsu et al., 1982 |
| Furumachi Pumice | FR | Northern Honshu | ST | 90 | Hayatsu et al., 1982 |
| Towada-H Tephra | To-H | Northern Honshu | $^{14}$C, OI | 15 | Machida and Arai 2003; Hayakawa, 1990; Arai et al., 1986 |
| Naruko-Yanagisawa | Nr-Y | Northern Honshu | $^{14}$C, OSL, FT | 41–63 | Machida and Arai, 2003 |
| Naruko-Nisaka Tephra | Nr-N | Northern Honshu | ST | 90 | Machida and Arai, 2003 |
| Dokusawa Tephra | DKS | Northern Honshu | ST | 90–100 | Matsuura et al., 2009 |
| Tanabu Tephra | Tn (A-C) | Northern Honshu | ST, OI | MIS 7–MIS 8 | Matsuura et al., 2014 |
| Ontake-1 Pumice | On-Pm1 | Central to northern Honshu | FT, K-Ar, ST | ca. 100 | Machida and Arai, 2003 |
| Sambe-Kisuki Tephra | SK | Across Honshu | ST | 110–115 | Toyokura et al., 1991 |
| Shimosueyoshi Loam | | In and Around Yokohama | FT | 120–130 | Toma, 1974 |
| Hakone Pumice Fall Deposit | Hk-KIP-7 | Chubu-Kanto (Central Japan) | ST | 130 | Suzuki, 1992 |
| Hakone Kissawa Pumice Layer | Hk-KIP-8 | Chubu-Kanto (Central Japan) | FT | 132 | Suzuki, 1992 |
| Miwa Lower Pumice Layer | Miwa-L | Chubu-Kanto (Central Japan) | ST | 130 | Suzuki, 1992 |
| Hakone-Tokyo Pumice | Hk-Tp | Around Tokyo | OSL | 67.5 ± 4.3 | Machida et al., 1987; Tsukamoto et al., 2010 |
| Matsue Tephra | DMP | Chugoku and Shikoku | ST | 110–120 | Inoue et al., 2005; Miura and Hayashi, 1991 |
| Aso-3 Tephra | Aso-3 | Central Kyushu - central Honshu | FT, K-Ar, ST | 120–135 | Machida and Arai, 2003 |
| Ata Tephra | Ata | In and around Japan | K-Ar, ST | 105–110 | Machida and Arai, 2003 |
| Aira-Tanzawa Tephra | AT | In and around Japan | $^{14}$C | 25.12 ± .27 | Machida 2002; Miyairi et al., 2004 |
| Aso-4 Tephra | Aso-4 | In and Around Japan | OI, K-Ar, ST | 87–89 | Takarada and Hoshizumi, 2020 |
| Kikai-Tanazawa tephra | K-Tz | In and Around Japan | ST, TL | 75–80 | Machida and Arai, 2003 |

\* OI = oxygen isotope dating, ST = stratigraphy correlation, FT = fission-track dating, $^{14}$C = carbon-14 dating, K-Ar = potassium-argon dating, OSL = optically stimulated luminescence dating, U = uranium thorium dating

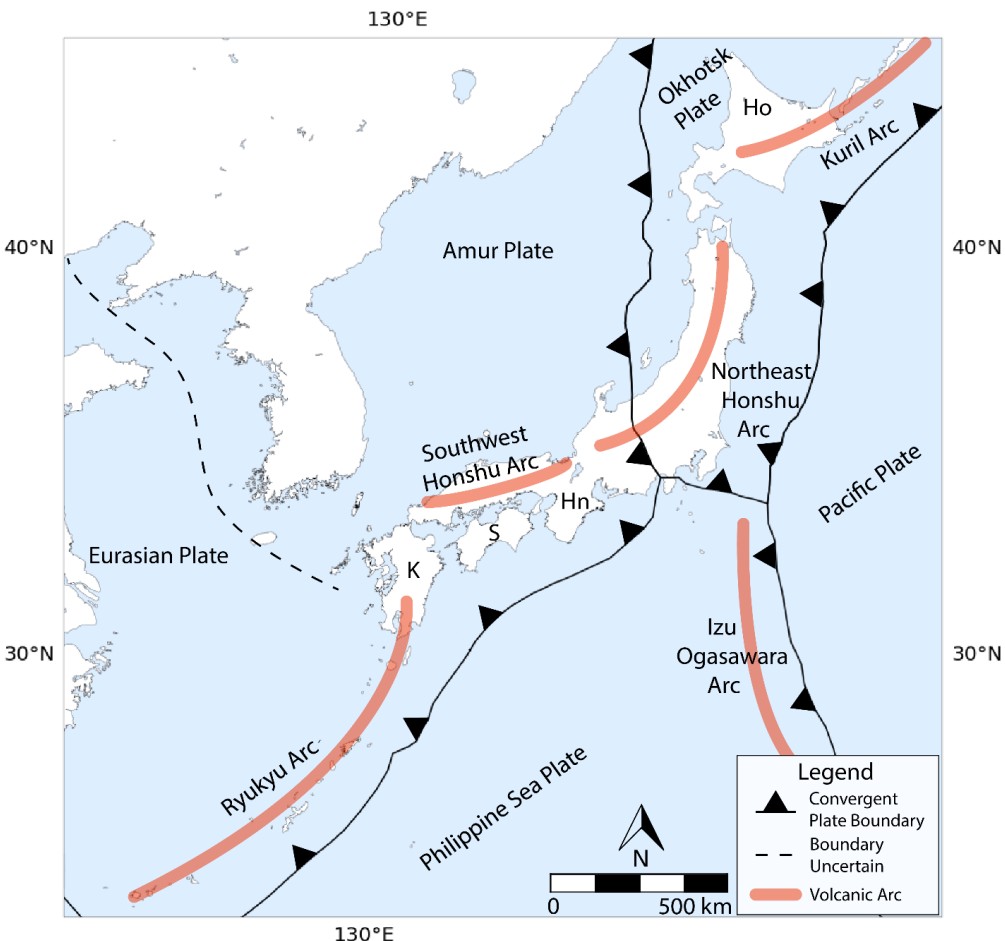

**Figure 1.** Modified from Taira (2001). An overview of the tectonic plates that compose and surround the Japanese Archipelago, detailing the interactions between the Okhotsk, Amura, Eurasian, Pacific, and Philippine Sea Plates, their plate boundaries, and the resulting volcanic arcs. Names of major islands of Japan are abbreviated: Ho = Hokkaido, Hn = Honshu, S = Shikoku, and K = Kyushu.



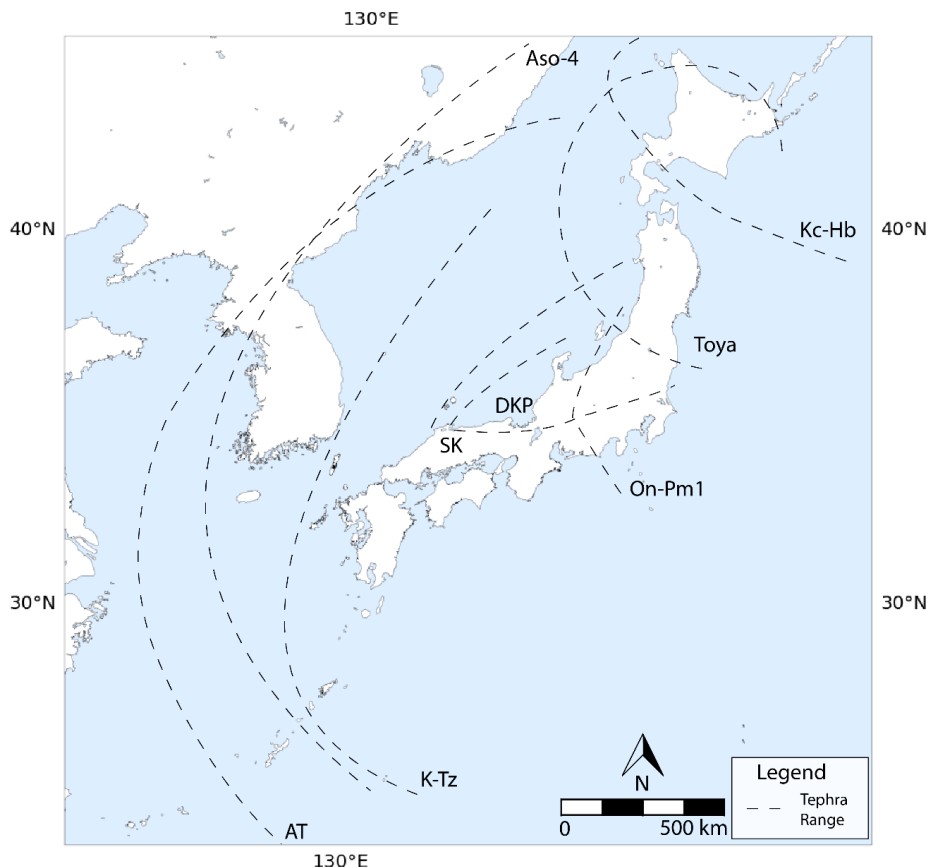

**Figure 2.** Tephra distribution map in and around Japan, modified from Machida (2002). Tephra recognized as key chronohorizons in this study include Toya, Kc-Hb, On-Pm1, Aso-4, K-Tz, SK, DKP, and AT.



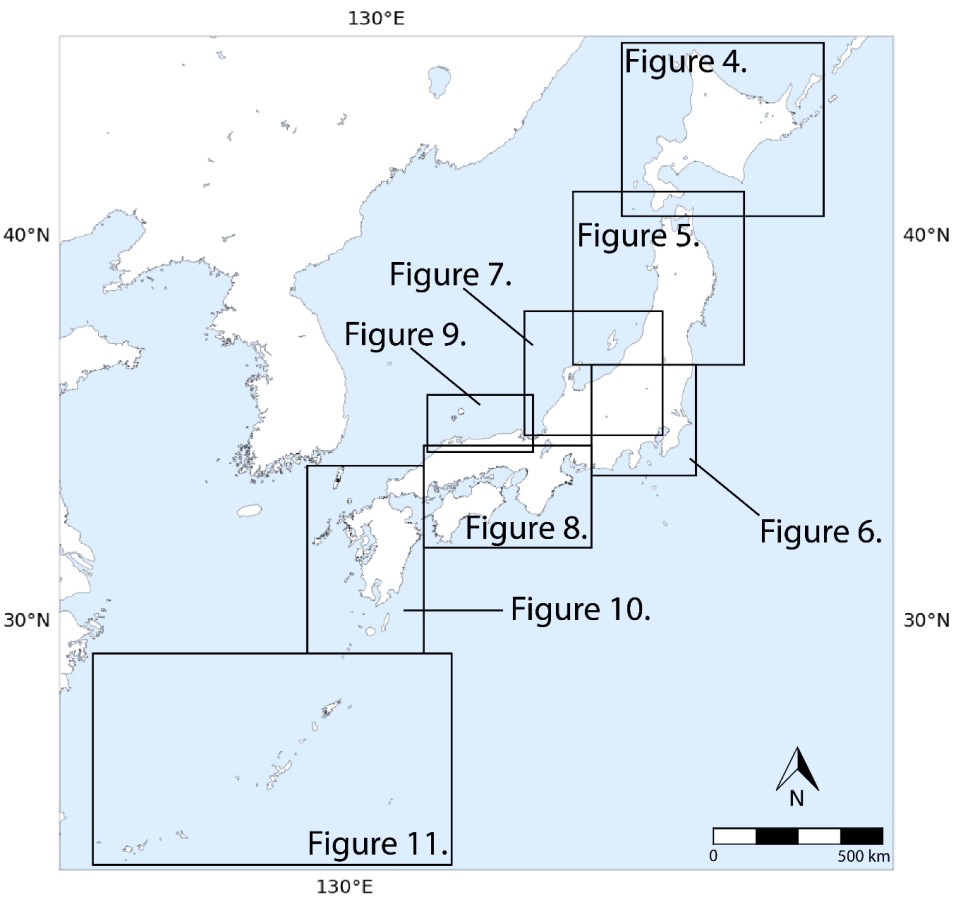

**Figure 3.** Map of Japan, indicating subsections in which MIS 5e sea-level proxies are examined.

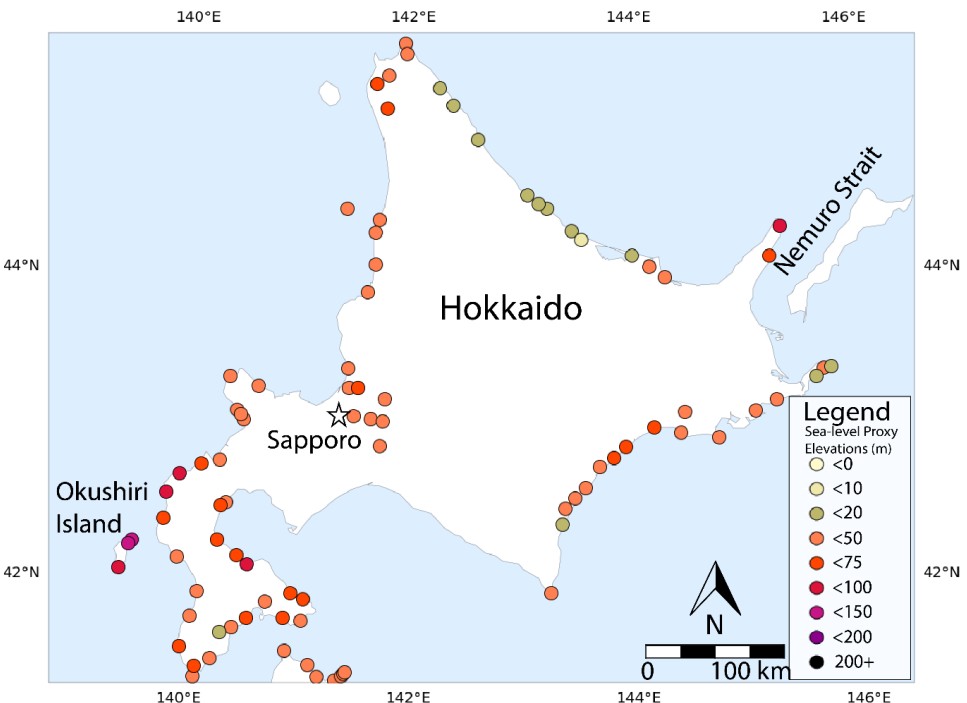

**Figure 4.** Sea-level elevation proxies in Hokkaido. Sea-level indicators (circles) elevation range indicated by color (see legend). Reference cities are indicated by stars.

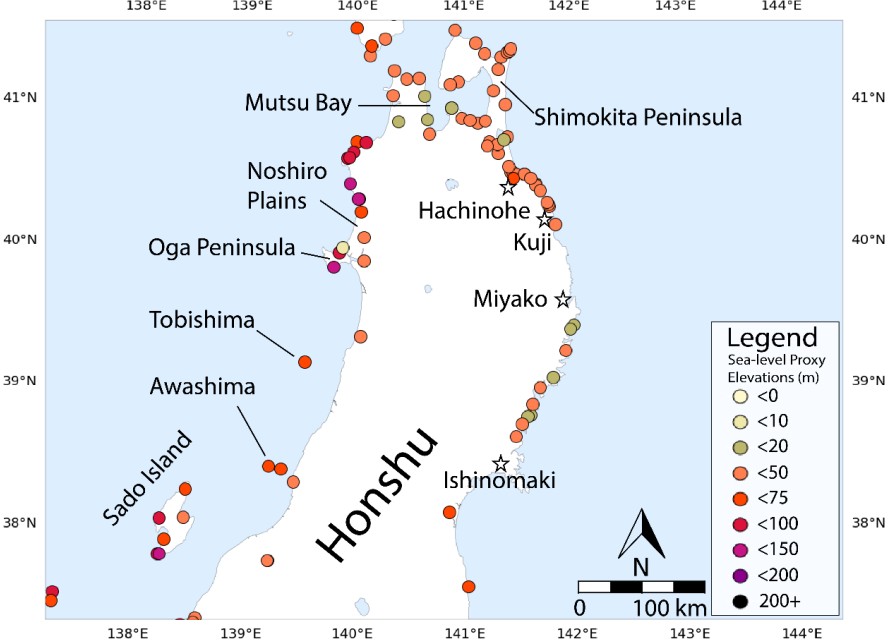

**Figure 5.** Sea-level elevation proxies in northern Honshu. Sea-level indicators (circles) elevation range indicated by color (see legend). Reference cities are indicated by stars.






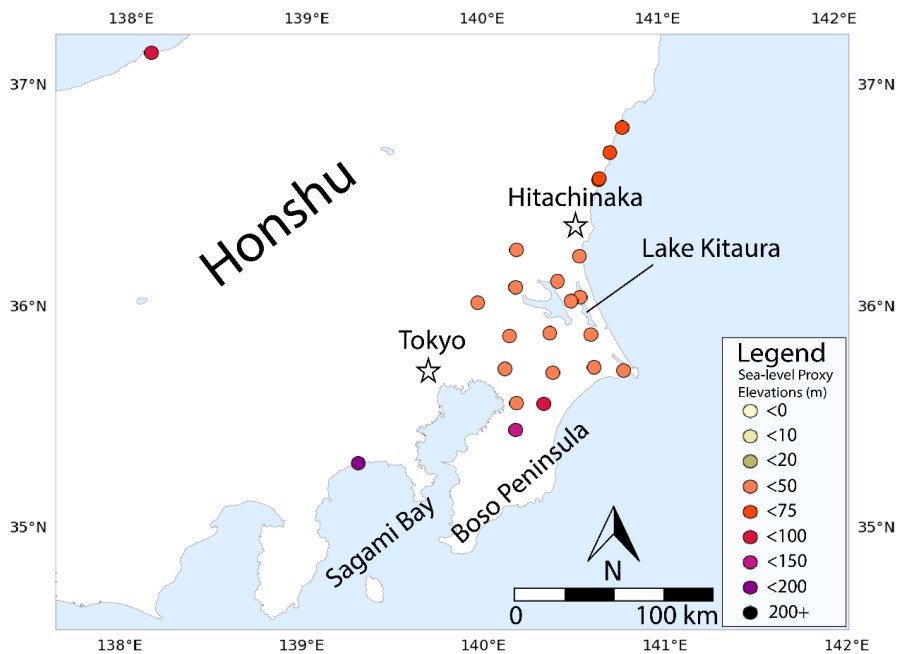

**Figure 6.** Sea-level elevation proxies in the Kanto region. Sea-level indicators (circles) elevation range indicated by color (see legend). Reference cities are indicated by stars.

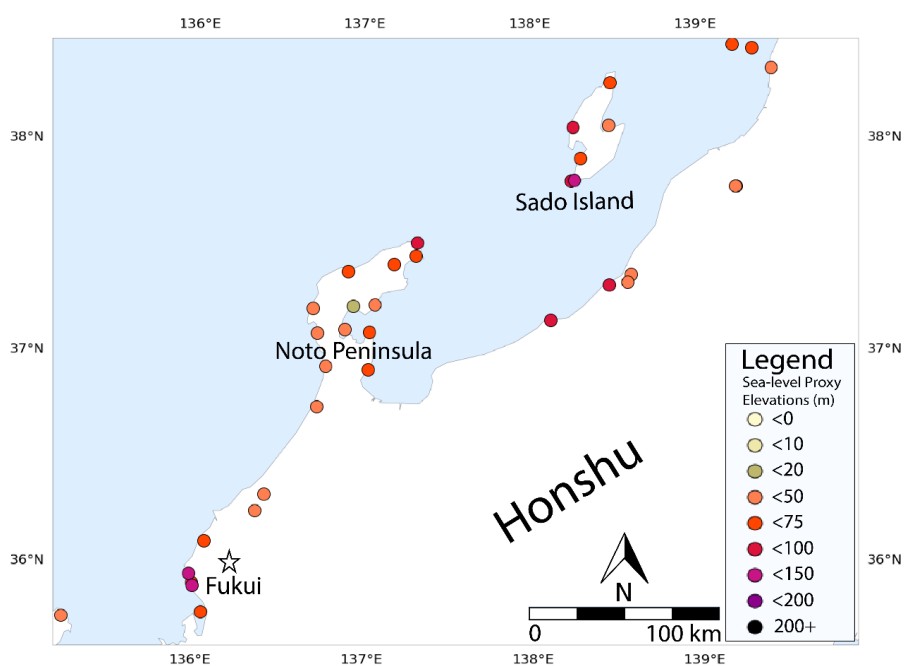

**Figure 7.** Sea-level elevation around the Noto Peninsula. Sea-level indicators (circles) elevation range indicated by color (see legend). Reference cities are indicated by stars.

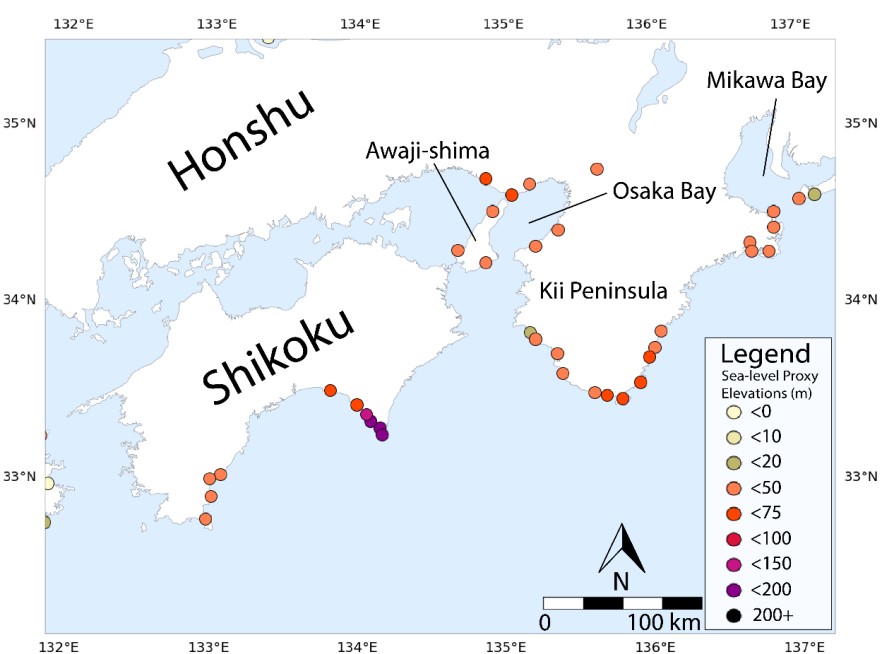

**Figure 8.** Sea-level elevation proxies in Shikoku and the Kii Peninsula. Sea-level indicators (circles) elevation range indicated by color (see legend).

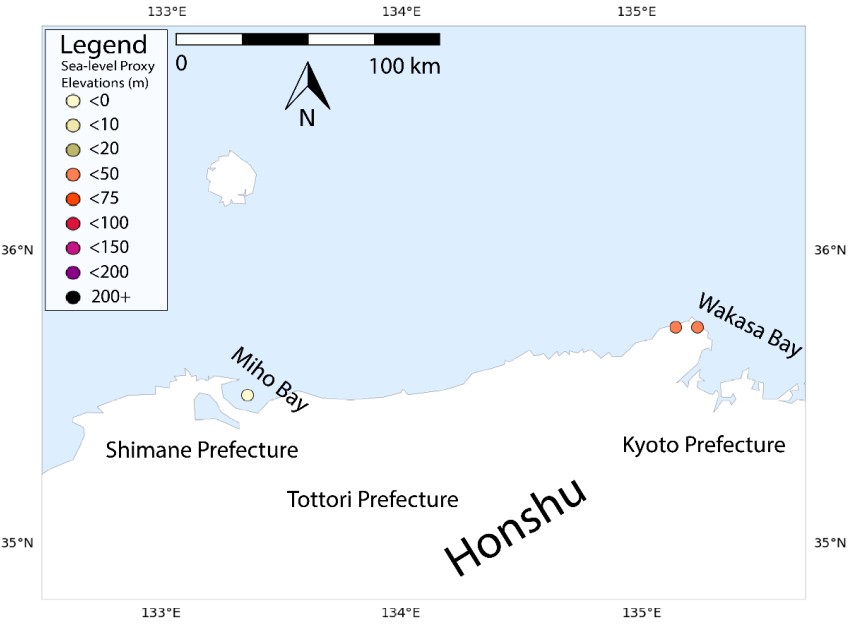

**Figure 9.** Sea-level elevation proxies along the Japan Sea (Kansai and Chugoku). Sea-level indicators (circles) elevation range indicated by color (see legend).

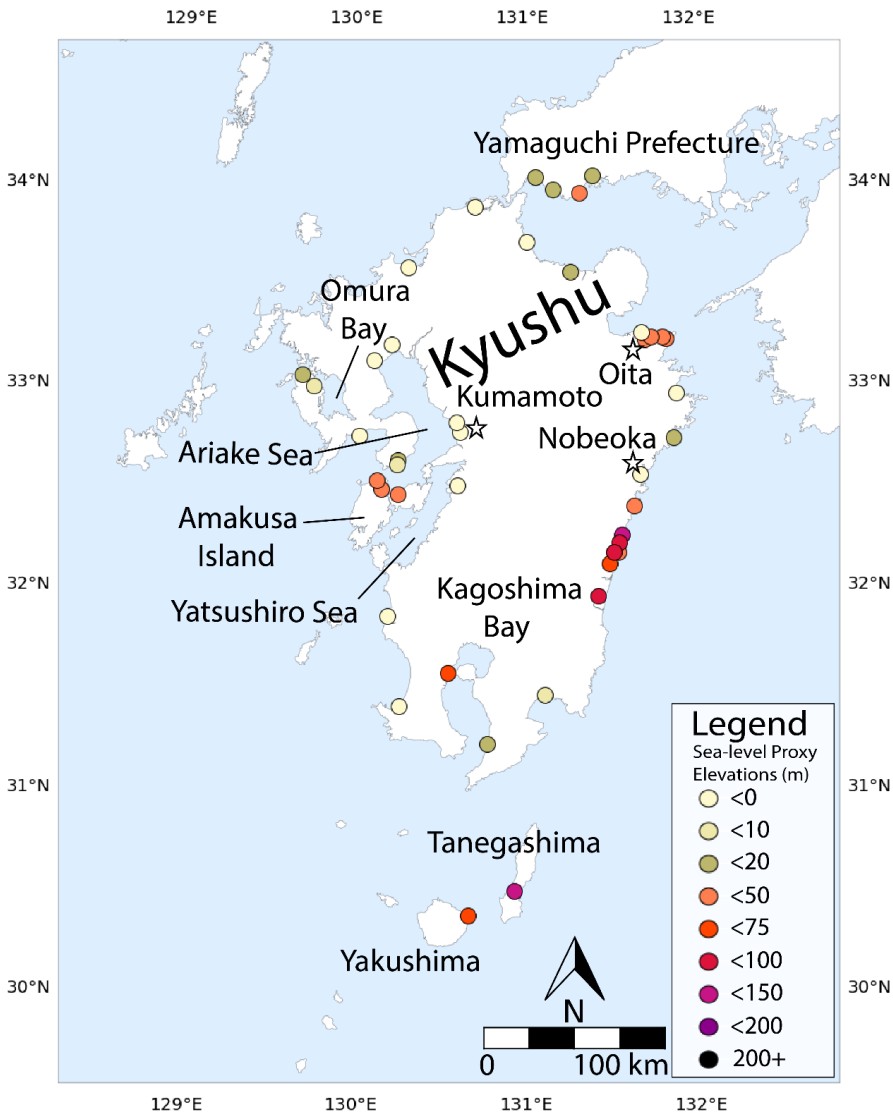

**Figure 10.** Sea-level elevation proxies in Kyushu and Yamaguchi. Sea-level indicators (circles) elevation range indicated by color (see legend). Reference cities are indicated by stars.






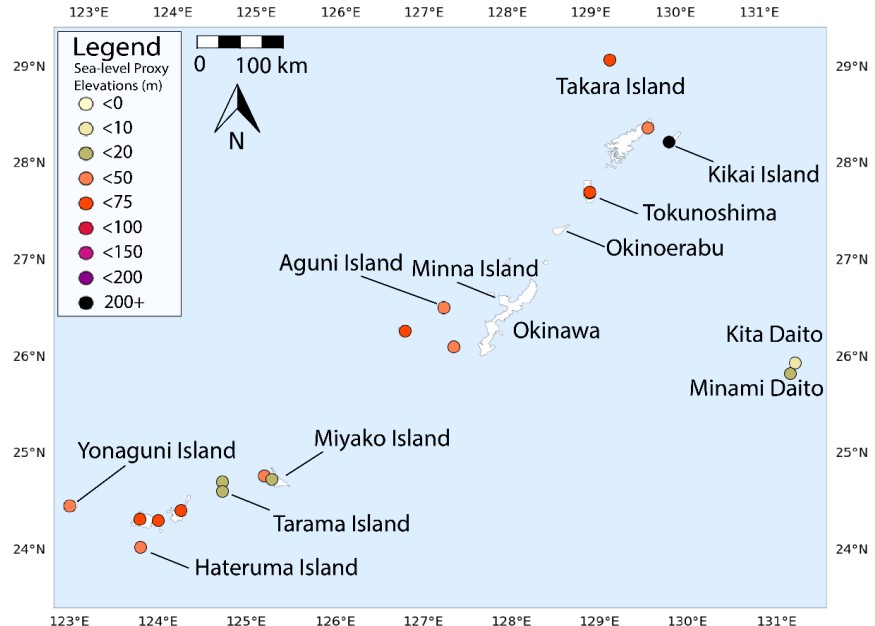

**Figure 11.** Sea-level elevation proxies in the Ryukyu Islands. Sea-level indicators (circles) elevation range indicated by color (see legend).