# Peer review of "A Review of MIS 5e Sea-level Proxies around Japan"

_Earth System Science Data, 2020_

## Referee Comment (RC1) · Thomas M. Cronin (Referee) · 23 Dec 2020

A Review of MIS 5e Sea-level Proxies around Japan By Evan Tam and Yusuke Yokoyama

General. This is an excellent paper, comprehensive, well-written and valuable in documenting last interglacial shorelines from extra-tropical regions in the Japanese archipelago. It's value lies not only in global MIS5e sea level reconstruction, but in documenting coastal uplift and subsidence given the very broad range of elevations of MIS5e paleoshorelines. The extensive tephra dating is a positive development. Having worked on Japanese Quaternary deposits myself years ago, I know the meticulous geomorphological and stratigraphic correlations are excellent, based on decades of

careful field work. The authors are careful to point out limitations of tephra chronology and paleo-sea level elevations from terraces. The maps are excellent. I wonder if it would be easy to add a map showing regions uplift and subsidence rates around Japan based on MMIS5e shorelines. The current maps show the actual elevations, but not rates. Otherwise, I cannot find any major flaws in the study. It will be an excellent addition to the WALIS program.

\Minor

Line 34 Otto-Bliesner Line 75 mollusks? Mollusca ? Line 504 coral-rich In Table 3 please double check the Genus name has upper case first letter but species names are lower case.

Please also note the supplement to this comment:
https://essd.copernicus.org/preprints/essd-2020-365/essd-2020-365-RC1-supplement.pdf

---

## Referee Comment (RC2) · Luigi Ferranti (Referee) · 13 Jan 2021

The paper by Tam and Yokoyama provides an overview of sites in Japan Archipelago coasts where evidence of indicators of the Last Interglacial shoreline were previously published. The compilation was taken in the frame of the WALIS project - the World Atlas of Last Interglacial Shorelines and follows its protocol as supporting databases show. To my knowledge a modern compilation of LIG data from Japan Archipelago is lacking so the effort is welcomed. The authors have subdivided the database description for the different islands and part of the islands where the LIG proxies show a more or less coherent elevation distribution. Large differences in elevations arise from the circumstantial position relative to active tectonic structures that causes uplift or subsidence. Overall, there is an exceptional age control on the terraces thanks to

the excellent correlation with dated tephrostratigraphic horizons and absolute dating. I found the compilation exhaustive and concise to the point.

Only at rare spots, I found some mistaken typing as follows: Line 67: add "on" before "marine" Line 117: Pleistocene upper case Line 133: amend "comparied" without the i Line 200: here the sentence is truncated Line 257-258: could you sjhow the location of these volcanic features on the map? Line 259: "sub-categories" – better subregions Line 264: "additionally ZP" – insert "by the" Line 336: Sagami Through – better Sagami trench subduction Line 343: unclear phrasing – maybe truncated sentence Line 367: "Elevations on the western side of the Kii peninsula generally increase in elevation" Line 369-370: "Terraces are described as one of a set of seven (H1–H4 and L1–L3, with L1 representing MIS 5e), and are described as..." Please avoid repetition.

Tables (bit please check the typing more carefully): Table 1: "sediments" – add "m" Tbale 2: "Amphitegina" – correct: Amphistegina

Luigi Ferranti

---

## Author Comment (AC1) · 2 Feb 2021

Dear Dr. Cronin,

Thank you for the generous review of the paper. Your comments have been a welcome addition, and will be included in the final version of the manuscript.

Specifically, we plan on adding a small section describing the uplift and subsidence rates around Japan based on our data. We have also made the minor edits mentioned in your review of this paper.

Answers to your comments are provided below. Thank you for your time and effort in this review.

[Figure]

Best wishes, Evan Tam, on the behalf of all authors

EDITOR'S COMMENTS Line 34 Otto-Bliesner Line 75 mollusks? Mollusca ? Word changed to Mollusca.

Line 504 coral-rich Edit made.

In Table 3 please double check the Genus name has upper case first letter but species names are lower case. Corrections have been made.

---

## Author Comment (AC2) · 2 Feb 2021

Dear Dr. Ferranti,

Thank you for the kind and extensive review of our paper. The comments you made were quite helpful in making this paper a smoother, more accurate review.

Answers to your comments are provided below. I have also added a revised Figure 2, which includes various tephra source volcanoes mentioned in the manuscript. Thank you for your time and effort in this review.

Best wishes, Evan Tam, on the behalf of all authors

EDITOR'S COMMENTS

Line 67: add "on" before "marine" Correction made.

Line 117: Pleistocene upper case Correction Made.

Line 133: amend "comparied" without the i Correction made.

Line 200: here the sentence is truncated Edit made.

Line 257-258: could you sjhow the location of these volcanic features on the map? Volcanoes mentioned in the manuscript have been added to Figure 2.

Line 259: "sub-categories" – better subregions Correction made.

Line 264: "additionally ZP" – insert "by the" Addition made.

Line 336: Sagami Through – better Sagami trench subduction Edit made.

Line 343: unclear phrasing – maybe truncated sentence Sentence changed to "did not report elevation values".

Line 367: "Elevations on the western side of the Kii peninsula generally increase in elevation" Sentence modified to "Elevations on the western side of the Kii Peninsula generally increase towards the southern tip. . .".

Line 369-370: "Terraces are described as one of a set of seven (H1–H4 and L1–L3, with L1 representing MIS 5e), and are described as. . ." Please avoid repetition. Changed to "Seven terrace levels are reported. . .and are described as. . .".

Tables (bit please check the typing more carefully): Table 1: "sedients" – add "m" Correction made.

Tbale 2: "Amphitegina" – correct: Amphistegina Correction made.
* * *
[Figure]

**Fig. 1.**

---

## Author Response (AR1)

Author's Response Outline

Below is a chronicle of all edits made based on reviews from Dr. Murray Wallace, Dr. Cronin, and Dr. Ferranti.

Correctional edits received from Colin Murray Wallace:

- The word in line 21 is 'with'
  Completed
- Line 15, perhaps the 'past 60 years' – hopefully they are not the last
  Done
- I feel that it would be good to indicate the key findings and general trends that you have noted in the data set towards the end of your abstract
  The following sentence has been added to the end of the abstract: "Sea-level proxy studies in Japan rely heavily on chronostratigraphic techniques and are recognized as reliable, though opportunities exist for further constraining through the further use of numerical age dating techniques."
- Line 37 s insteas of c for isostasy and eustasy
  Done
- Line 37 These data are
  Done
- Line 67 information on marine terrace
  Done
- Line 91 Machida (1975) used tephrochronology … with fission track ages to correlate high sea-level stages
  Done
- Line 105 heavily used in
  Done
- Line 117 Late Pleistocene?
  Capitalization added
- Line 132 not sure what you mean by 'updated tephra'?
  Wording changed to "Updated ages of tephra defined marine terraces"
- Line 133 compared with the original
  Done
- Line 134 Large uncertainty  (an error, strictly speaking is something that is wrong)
  Done
- Line 150 and other instances – please avoid the term 'absolute' in geochronology – numeric dating (for rationale, please see Colman 1987 Quaternary Research, 28, 314-319).
  Changes made here and in other instances. Insightful edit! Changed to "numerical age dating"
- Line 157 uncertainties instead of errors
  Done
- Line 161 for elegant variation, please remove one of the instances of 'techniques' – perhaps 'geochronological methods' could replace the first instance?
  Substitution made
- Line 163 perhaps mention in passing the reason for the limited number of OSL-based investigations.
  Added the word investigative to underline the use of the technique in Japan is still in its infancy
- Line 213 and all other instances – I would suggest joining 'paleo' with 'sea' to form one word, as paleo (palaeo) of itself is not a word
  Changes made
- Line 256 lower case k for ka (upper case K refers to Kelvin)
  Changes Made
- Line 284 What does 'proxy elevation' mean? Please clarify
  Changed to "Recorded elevations of sea-level proxies"

- Line 298 Peninsula
  Correction made
- Line 304 OSL ages (not dates which are unique calendar events)
  Corrected
- Line 318 word choice – perhaps replace 'drastically' with 'significantly' ? and in line 350
  Substitutions made
- Line 353 insert 'I' stratigraphically
  Change made from stratographic to stratigraphic
- Line 371 Osaka Bay – and for other instances where the word is used as a formal noun
  Correction made
- Line 375 range 'widely' rather than 'very greatly'
  Substitution made
- Lines 378-379 correlation to constrain ages, though
  Edit made
- Line 381 Five terrace layers
  Change made
- Line382 amino acid racemization
  Edit made
- Line 400 metres – not sure if you prefer to retain American spelling.
  American spelling sustained.
- Line 440 considered reliable at what level – I feel that a qualification is needed here
  "scientifically" added as a qualifier
- Line 458 numeric dating
  Changed to numerical age dating, see edit for line 150
- Line 464 mollusc ?
  Edit made
- Line 472 not included in the database
  Edit made
- Line 474 for the entire Japanese Archipelago
  Changes made
- Line 477 has not been emphasized
  Edit made
- Line 479 numeric dating and in Line 481
  Changed to numerical age dating, see edit for line 150
- Line 484 Numerous studies have … to correlate with other …
  Edits made
- Line 489 or interglacial sea levels
  Edit made
- Line 499 molluscs?
  Edit made
- Line 509 numeric dating
  Changed to numerical age dating, see edit for line 150
- Table 1 – in the column Proxy Description 'accumulation of sediments'
  Correction made
- Additional section discussing Rates of Uplift in Japan added

Correctional edits Received from Thomas Cronin:

EDITOR'S COMMENTS

- Line 34 Otto-Bliesner

Correction made

- Line 75 mollusks? Mollusca ?
  Word changed to Mollusca.
- Line 504 coral-rich
  Edit made.
- In Table 3 please double check the Genus name has upper case first letter but species names are lower case.
  Corrections have been made.

Correctional edits received from Luigi Ferranti:

EDITOR'S COMMENTS

- Line 67: add "on" before "marine"
  Correction made.
- Line 117: Pleistocene upper case
  Correction Made.
- Line 133: amend "comparied" without the i
  Correction made.
- Line 200: here the sentence is truncated
  Edit made.
- Line 257-258: could you sjhow the location of these volcanic features on the map?
  Volcanoes mentioned in the manuscript have been added to Figure 2.
- Line 259: "sub-categories" – better subregions
  Correction made.
- Line 264: "additionally ZP" – insert "by the"
  Addition made.
- Line 336: Sagami Through – better Sagami trench subduction
  Edit made.
- Line 343: unclear phrasing – maybe truncated sentence
  Sentence changed to "did not report elevation values".
- Line 367: "Elevations on the western side of the Kii peninsula generally increase in elevation"
  Sentence modified to "Elevations on the western side of the Kii Peninsula generally increase towards the southern tip…".
- Line 369-370: "Terraces are described as one of a set of seven (H1–H4 and L1–L3, with L1 representing MIS 5e), and are described as. . ." Please avoid repetition.
  Changed to "Seven terrace levels are reported…and are described as…".
- Tables (bit please check the typing more carefully): Table 1: "sedients" – add "m"
  Correction made.
- Table 2: "Amphitegina" – correct: Amphistegina
  Correction made.